# A ubiquitin-like protein encoded by the "noncoding" RNA TINCR promotes keratinocyte proliferation and wound healing

**Akihiro Nita[1], Akinobu Matsumoto[1]\*, Ronghao Tang[1], Chisa Shiraishi[1], Kazuya Ichihara[1], Daisuke Saito[2], Mikita Suyama[2], Tomoharu Yasuda[3], Gaku Tsuji[4], Masutaka Furue[4], Bumpei Katayama[5], Toshiyuki Ozawa[5], Teruasa Murata[6], Teruki Dainichi[7], Kenji Kabashima[6], Atsushi Hatano[8], Masaki Matsumoto[8], Keiichi I. Nakayama[1]\***

1 Department of Molecular and Cellular Biology, Medical Institute of Bioregulation, Kyushu University, Fukuoka, Japan, 2 Division of Bioinformatics, Medical Institute of Bioregulation, Kyushu University, Fukuoka, Japan, 3 Department of Immunology, Graduate School of Biomedical and Health Sciences, Hiroshima University, Hiroshima, Japan, 4 Department of Dermatology, Graduate School of Medical Sciences, Kyushu University, Fukuoka, Japan, 5 Department of Dermatology, Osaka City University Graduate School of Medicine, Osaka, Japan, 6 Department of Dermatology, Kyoto University Graduate School of Medicine, Kyoto, Japan, 7 Department of Dermatology, Kagawa University Faculty of Medicine, Kagawa, Japan, 8 Department of Omics and Systems Biology, Niigata University Graduate School of Medical and Dental Sciences, Niigata, Japan

\* akinobu@bioreg.kyushu-u.ac.jp (AM); nakayak1@bioreg.kyushu-u.ac.jp (KIN)

**Data Availability Statement:** The RNA-seq data generated during this study have been deposited in the DDBJ database under accession numbers DRA011760 (mouse epidermis) and DRA011761

## Abstract

Although long noncoding RNAs (lncRNAs) are transcripts that do not encode proteins by definition, some lncRNAs actually contain small open reading frames that are translated. TINCR (terminal differentiation–induced ncRNA) has been recognized as a lncRNA that contributes to keratinocyte differentiation. However, we here show that TINCR encodes a ubiquitin-like protein that is well conserved among species and whose expression was confirmed by the generation of mice harboring a FLAG epitope tag sequence in the endogenous open reading frame as well as by targeted proteomics. Forced expression of this protein promoted cell cycle progression in normal human epidermal keratinocytes, and mice lacking this protein manifested a delay in skin wound healing associated with attenuated cell cycle progression in keratinocytes. We termed this protein TINCR-encoded ubiquitin-like protein (TUBL), and our results reveal a role for TINCR in the regulation of keratinocyte proliferation and skin regeneration that is dependent on TUBL.

## Author summary

Although, by definition, long noncoding RNAs (lncRNAs) are transcripts that do not encode proteins, recent studies have shown that some lncRNAs actually contain small open reading frames (ORFs) that are translated. Although TINCR (terminal differentiation–induced ncRNA) was originally identified as a lncRNA that contributes to keratinocyte differentiation, recent mass spectrometry–based analysis has suggested that TINCR is

(NHEKs). The MS data have been deposited with the ProteomeXchange Consortium (http://proteomecentral.proteomexchange.org) via the JPOST partner repository under the data set identifiers PXD026388 (HaCaT cells), PXD026389 (NHEKs), and PXD026392 (mouse epidermis).

**Funding:** This work was supported in part by KAKENHI grants from Japan Society for the Promotion of Science (JSPS) and the Ministry of Education, Culture, Sports, Science, and Technology of Japan (https://www.jsps.go.jp/english/) to A.M. (20H05928) and to K.I.N. (18H05215). The funders had no role in study design, data collection and analysis, decision to publish, or preparation of the manuscript.

**Competing interests:** The authors have declared that no competing interests exist.

translated. Translation of the ORF within the TINCR lncRNA was validated by generating mice that harbor an in-frame insertion of the FLAG epitope tag sequence at the COOH-terminus of the ORF. The TINCR-encoded protein contains a ubiquitin-like (Ubl) domain and was designated TUBL (TINCR-encoded ubiquitin-like protein). TUBL accelerated cell cycle progression in keratinocytes, and TUBL-deficient mice manifested delayed wound healing after injury with a biopsy punch as a result of attenuated keratinocyte proliferation. TUBL plays a key role at the protein level in the maintenance of skin homeostasis after injury through promotion of keratinocyte proliferation.

## Introduction

Skin has a barrier function that protects it from dehydration, wounding, ultraviolet radiation, and the entry of microbes. The outermost layer of skin, the epidermis, consists of a stratified squamous epithelium of keratinocytes delimited by a basal membrane, but it also contains melanocytes as well as Langerhans and Merkel cells [1,2]. A key property of skin is its high self-renewal capacity. During the wound healing process of skin, the efficient proliferation of epithelial keratinocytes is essential for re-epithelialization of the wound and restoration of barrier function [3–5], with many factors having been shown to contribute to keratinocyte proliferation and skin regeneration [6–8].

TINCR (terminal differentiation–induced noncoding RNA) was originally identified as a long noncoding RNA (lncRNA) that serves as a positive regulator of human epidermal differentiation [9]. TINCR binds to the protein STAU1 as well as to a series of mRNAs required for keratinocyte differentiation. These mRNAs interact with TINCR via a 25-nucleotide (nt) motif known as the TINCR box and appear to be stabilized by the TINCR-STAU1 complex. Recent studies have also revealed a role for TINCR in cancer, including breast, lung, liver, esophageal, colon, bladder, prostate, gastric, and oral tumors [10–21]. Forced expression of TINCR promotes cell proliferation and metastasis through activation of the Wnt/β-catenin signaling pathway in oral squamous cell carcinoma [20]. In addition, TINCR expression is up-regulated in hepatocellular carcinoma, resulting in activation of the phosphoinositide 3-kinase (PI3K)–Akt–mechanistic target of rapamycin (mTOR) signaling pathway through sequestration of the microRNA miR-7-5p [21]. These findings implicate TINCR in processes that contribute to the development and progression of cancer, such as regulation of the cell cycle.

Expression of lncRNAs is often restricted to a limited number of tissues, such as brain, testis, blood, liver, and skin [22]. These RNAs have been implicated in a wide array of cellular processes, including transcriptional regulation as well as cell differentiation and reprogramming [23]. Similar to mRNAs, many lncRNAs are transcribed by RNA polymerase II and then spliced, capped, and polyadenylated [24]. We and others have recently shown that some lncRNAs actually encode small functional proteins [25–28], with recent mass spectrometry (MS) analysis also having provided evidence for translation of TINCR [29]. Given that the data dependent acquisition (DDA) mode of proteomics analysis is associated with a certain number of false positive identifications, however, further investigations with different approaches are required to verify the endogenous expression of this putative TINCR-encoded protein.

We have now generated mice that harbor an in-frame knock-in of the FLAG epitope tag at the COOH-terminus of the open reading frame (ORF) in TINCR and confirmed translation of the ORF. The encoded protein contains a ubiquitin-like (Ubl) domain, and its forced expression promoted cell cycle progression in normal human epidermal keratinocytes (NHEKs). We therefore named this protein TINCR-encoded ubiquitin-like protein (TUBL). We also

generated mice with a 1-bp deletion that abrogates expression of TUBL as a result of a frame-shift without affecting the secondary structure of TINCR. The TUBL-deficient mice manifested delayed wound healing after injury with a biopsy punch in association with attenuation of keratinocyte proliferation. Our results thus indicate that TINCR encodes the ubiquitin-like protein TUBL and contributes to skin homeostasis after injury by regulating cell cycle progression in keratinocytes through this protein.

## Results

### TINCR encodes the ubiquitin-like protein TUBL

PhyloCSF is a comparative genomics method to identify evolutionarily conserved ORFs in silico on the basis of the alignment of nucleotide sequences from multiple species [30]. We applied this method to many putative lncRNAs in order to identify hidden small proteins encoded by short ORFs. This analysis revealed TINCR to contain a potential ORF, given the consecutive positive signals of PhyloCSF observed for both the human and mouse sequences (Fig 1A). The predicted protein consists of 87 amino acids in human and is conserved across amphibians (*Xenopus tropicalis*), reptiles (*Anolis carolinensis*), and birds (*Gallus gallus*) as well as mammals (*Homo sapiens*, *Mus musculus*) (Fig 1B), although no apparent corresponding orthologs are present in fish or insects. The amino acid sequence of this protein shows a high level of similarity to that of ubiquitin (Fig 1C). Hereafter, we refer to this predicted protein encoded by the small ORF of TINCR as TINCR-encoded ubiquitin-like protein (TUBL).

To validate translation from this ORF, we transiently transfected HEK293T cells with an expression construct for mouse TINCR with a FLAG epitope tag sequence inserted at the COOH-terminus of the ORF. Immunoblot analysis indeed confirmed the expression of FLAG-tagged TUBL in these cells (Fig 1D). Human TINCR contains one potential AUG initiation codon, whereas two potential such codons are present in mouse TINCR (Fig 1B). We therefore generated constructs in which one or both AUG codons of mouse TINCR were deleted in-frame and expressed them in HEK293T cells. Expression of TUBL was detected when only one AUG codon was deleted but not when both were (Fig 1D), indicating that both codons are capable of initiating translation. Immunofluorescence microscopic analysis revealed that FLAG epitope–tagged TUBL was diffusely distributed throughout the cytoplasm when expressed in HeLa cells (Fig 1E).

Ubiquitin-like proteins are divided into two families, type 1 and type 2 [31]. Type 1 Ubl proteins contain a diglycine motif at the COOH-terminus and are covalently attached to other proteins through the action of an enzymatic cascade similar to that for ubiquitin, whereas type 2 Ubl proteins lack this motif and are not ligated to other proteins. Indeed, immunoblot analysis with antibodies to the V5 epitope tag revealed that V5-tagged NEDD8, a type 1 Ubl protein, was conjugated to other proteins in HEK293T cells, whereas a mutant form of NEDD8 lacking the diglycine motif was not (Fig 1F). The amino acid sequence of TUBL does not contain a diglycine motif (Fig 1B), and neither human nor mouse TUBL tagged with the V5 epitope was found to be conjugated to other proteins (Fig 1F), indicating that TUBL is type 2 Ubl protein. Addition of the V5 tag at the NH$_2$-terminus appeared to impair expression of mouse TUBL but not that of human TUBL (Fig 1F). As mentioned above, unlike that of human or other species, mouse TUBL contains two potential AUG initiation codons (Fig 1B). We inserted the V5 tag immediately after the second AUG initiation codon for mouse TUBL (Fig 1F), which might have hindered translation initiation. It is also possible that addition of the V5 tag at the NH$_2$-terminus resulted in instability specifically of mouse TUBL by an unknown mechanism.

To examine endogenous expression of TUBL, we generated knock-in mice that harbor a FLAG epitope tag sequence at the COOH-terminus of the protein. Lysates prepared from

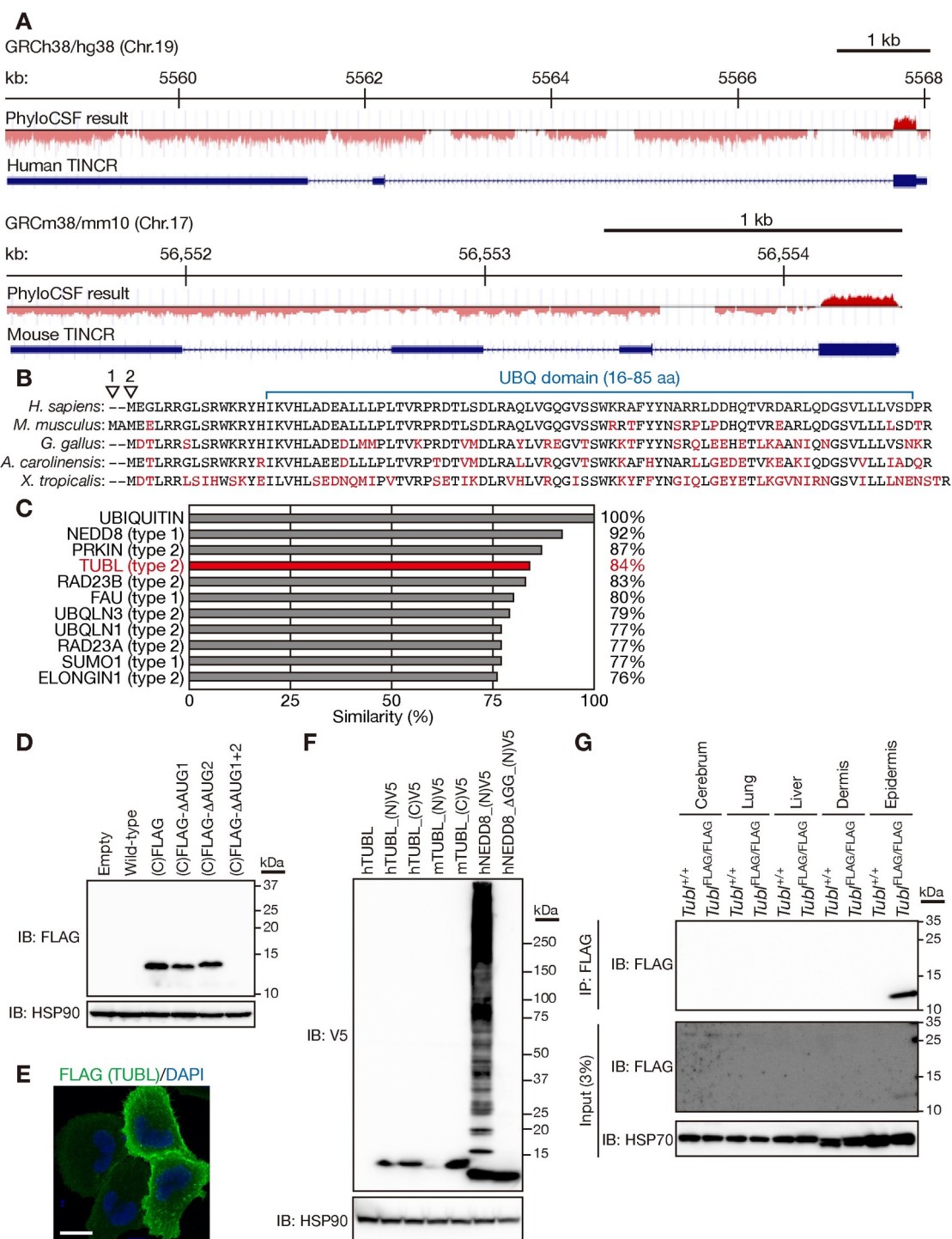

**Fig 1. TINCR encodes the ubiquitin-like protein TUBL.** (A) Smoothed PhyloCSF peaks for human and mouse TINCR. (B) Comparison of the predicted amino acid sequence of human TUBL with that of the mouse, avian, reptile, and amphibian proteins. Arrowheads indicate the first and second methionines of mouse TUBL. Amino acids that differ between species are shown in red. (C) Amino acid sequence similarity for human TUBL and other human ubiquitin-like proteins compared with human ubiquitin. (D) Immunoblot (IB) analysis of HEK293T cells transiently transfected with expression vectors for mouse TINCR with a FLAG epitope tag sequence inserted at the COOH-terminus (C) of the TUBL ORF or for its ΔAUG mutants. The analysis was performed with M2 antibodies to FLAG and antibodies to HSP90 (loading control). (E) Immunofluorescence analysis of FLAG in HeLa cells expressing FLAG epitope–tagged mouse TUBL. Nuclei were stained with 4′,6-diamidino-2-phenylindole (DAPI). Scale bar, 20 μm. (F) Immunoblot analysis of HEK293T cells transiently transfected with expression vectors for human (h) or mouse (m) TUBL with

or without an $NH_2$ (N)–or COOH (C)–terminal V5 epitope tag or for human NEDD8 with an $NH_2$-terminal V5 tag or its mutant (ΔGG) lacking the diglycine motif. (G) Lysates of tissues from wild-type ($Tubl^{+/+}$) mice or mice harboring a FLAG epitope tag sequence at the COOH-terminus of the TUBL ORF ($Tubl^{FLAG/FLAG}$ mice) were subjected to immunoprecipitation (IP) with antibodies to FLAG, and the resulting precipitates as well as the original lysates (3% of input for immunoprecipitation) were subjected to immunoblot analysis with antibodies to FLAG and to HSP70 (loading control).

mouse epidermis were subjected to immunoprecipitation with antibodies to FLAG, and the resulting precipitates were subjected to immunoblot analysis with the same antibodies. Such analysis confirmed the endogenous expression of TUBL in epidermis (Fig 1G), indicating that TINCR indeed encodes a Ubl domain–containing protein, TUBL.

## TUBL promotes cell cycle progression in human keratinocytes

To investigate the role of TUBL in keratinocytes, we infected NHEKs with a lentivirus encoding human TUBL (including only the coding region of TINCR, without the 5' and 3' untranslated regions) or with a virus encoding green fluorescent protein (GFP) as a control, and we then examined the gene expression profiles of these cells. RNA-sequencing (RNA-seq) analysis identified 368 differentially expressed genes: 36 up-regulated and 332 down-regulated genes with a $log_2$[fold change] of $>0.5$ or $<-0.5$ and adjusted p value $<0.05$ in the cells expressing TUBL compared with the control cells (S1 Table). Gene set enrichment analysis (GSEA) revealed that cell cycle–related gene sets were substantially up-regulated and gene sets specific for keratinocyte differentiation were significantly down-regulated in the cells expressing TUBL (Fig 2A–2C).

To validate the effect of TUBL on the expression of cell cycle–related genes, we examined cell cycle kinetics with a 5-bromo-2'-deoxyuridine (BrdU) incorporation assay. The proportion of cells in S phase of the cell cycle (BrdU$^+$ fraction) was markedly increased for NHEKs expressing TUBL compared with control cells (Fig 2D). Furthermore, RNA-seq analysis revealed that the expression of early or late differentiation marker genes was down-regulated in cells expressing TUBL (Fig 2E). These results suggested that TUBL promotes the proliferation and attenuates the differentiation of human keratinocytes.

Whereas we found that TUBL suppressed keratinocyte differentiation, TINCR was previously shown to promote such differentiation [9]. We therefore performed a calcium-induced in vitro differentiation assay for NHEKs transfected with the small interfering RNA (siRNA) used in this previous study (Lincode) [9] or with two additional siRNAs (Silencer Select). Reverse transcription (RT) and quantitative polymerase chain reaction (qPCR) analysis confirmed that all of the siRNAs substantially depleted TINCR (Fig 2F and 2G). However, whereas depletion of TINCR with the siRNA used in the previous study suppressed differentiation, that with the other two siRNAs did not affect the differentiation status of the cells as evaluated on the basis of expression of the gene for the late differentiation marker involucrin (Fig 2F). These results suggested that the inhibitory effect of TINCR depletion on differentiation apparent in the previous study might have been attributable to an off-target effect. In addition, we found that the proportion of cells in S phase of the cell cycle was markedly decreased for NHEKs transfected with the siRNAs targeting both TINCR and TUBL compared with those transfected with control siRNAs (Fig 2H), consistent with our results for NHEKs stably expressing TUBL (Fig 2D).

## TUBL, rather than TINCR, is essential for promotion of the keratinocyte cell cycle

Although expression of only the TUBL ORF region of TINCR promoted cell cycle progression in keratinocytes, we could not formally exclude the possibility that the corresponding RNA

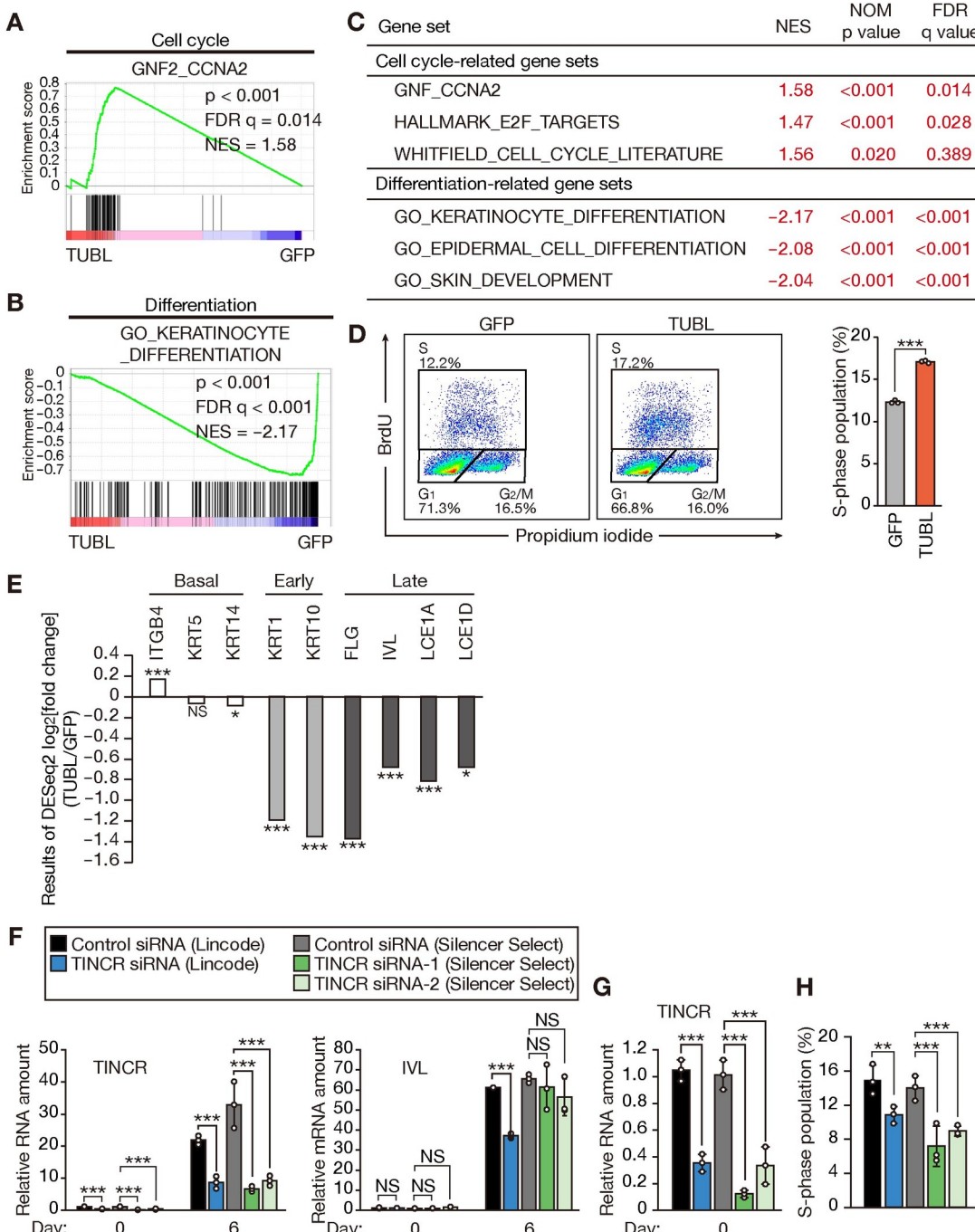

**Fig 2. TUBL expression promotes cell cycle progression and inhibits differentiation in human keratinocytes.** (A, B) GSEA plots for gene sets related to the cell cycle (A) or keratinocyte differentiation (B) constructed from RNA-seq data obtained for NHEKs expressing GFP or human TUBL. FDR, false discovery rate; NES, normalized enrichment score. (C) Summary of GSEA results for gene sets related to the cell cycle or keratinocyte differentiation. NOM, nominal. (D) Flow cytometric traces and quantification of BrdU incorporation for NHEKs stably expressing GFP or TUBL. Data in the right panel are means ± SD (n = 3 independent experiments). ***p < 0.005 (Student's t test). (E) RNA-seq results for differentially expressed genes related to basal, early, or late keratinocyte differentiation. The results for NHEKs expressing TUBL were normalized by those for NHEKs expressing GFP. *p < 0.05, ***p < 0.005; NS, not significant (adjusted p values). (F, G) RT-qPCR analysis of TINCR and involucrin (IVL) mRNA abundance in NHEKs transfected with the indicated control or TINCR siRNAs for 1 day and then subjected to calcium-induced differentiation for 0 or 6 days (F). The results for TINCR abundance at day 0 of differentiation are expanded in (G). Data are means ± SD (n = 3 independent experiments). ***p < 0.005, NS (Student's t test). (H) Quantification of BrdU incorporation for NHEKs transfected with the indicated control or TINCR siRNAs for 2 days. Data are means ± SD (n = 3 independent experiments). **p < 0.01, ***p < 0.005 (Student's t test).

component is responsible for this effect. To demonstrate that the TUBL protein is indeed functional, we introduced a 1-bp deletion (del) in the ORF that induces a frameshift and thereby abrogates TUBL expression while likely preserving the secondary structure of the RNA (Fig 3A and 3B). We also introduced synonymous mutations (SM) into the ORF that preserve the TUBL amino acid sequence but likely substantially alter the secondary structure of TINCR. Furthermore, we generated an ORF containing both the 1-bp deletion and synonymous mutations, thereby disrupting both protein expression and RNA structure. We then infected mouse primary keratinocytes with retroviruses containing either the wild-type (WT) ORF or one of the three mutant constructs. Expression of WT or SM forms of TINCR resulted in a marked increase in the percentage of cells in S phase of the cell cycle, whereas expression of WT_del or SM_del forms had no such effect (Fig 3C and 3D).

Forced expression of WT or SM forms of TINCR also moderately attenuated expression of the involucrin gene before the induction of differentiation in vitro, whereas such an effect was not observed at 6 days after the onset of differentiation induction (Fig 3E), suggestive of a weak or indirect inhibitory effect of TUBL on keratinocyte differentiation. As was apparent in NHEKs (Fig 2F), marked up-regulation of TINCR was observed in association with differentiation in control primary mouse keratinocytes as well as in cells expressing WT_del or SM_del mutants (Fig 3E). However, this effect was greatly suppressed in cells expressing WT or SM forms of TINCR (Fig 3E), suggestive of a negative feedback mechanism by which TUBL suppresses the induction of TINCR during keratinocyte differentiation. To investigate whether TINCR might also affect the expression of TUBL, we transiently transfected HEK293T cells with expression constructs both for TUBL with a COOH-terminal FLAG tag (ORF region only) and for an almost full-length form of TINCR with a 1-bp deletion that disrupts the TUBL ORF (WT_full_del). We found that the expression level of FLAG-tagged TUBL was not affected by the presence of the WT_full_del form of TINCR (Fig 3F), suggesting that TINCR is unlikely to contribute to the regulation of TUBL expression. Collectively, these results indicated that TUBL protein, but not the secondary structure of TINCR, is required for function in the regulation of cell cycle progression.

## TINCR is abundant in proliferating keratinocytes

Given that the abundance of TINCR was up-regulated during keratinocyte differentiation (Figs 2F and 3E), it might have been expected that TINCR promotes such differentiation. However, we found that overexpression of TUBL promoted keratinocyte proliferation in association with inhibition of differentiation (Figs 2 and 3A–3E). We therefore examined the tissue distribution of TINCR, and found that it was expressed not only in differentiated keratinocytes but also in actively proliferating primary mouse keratinocytes to a greater extent than in other adult mouse tissues (Fig 3G), suggesting that TUBL contributes to cell cycle progression in proliferating keratinocytes. However, given that TINCR was expressed at a higher level in epidermis than in proliferating keratinocytes, TUBL may also function in terminally differentiated keratinocytes, such as in formation of the cornified envelope (see Discussion).

## Generation of TUBL-deficient mice

We next generated mice deficient in TUBL as a result of the introduction of a 1-bp deletion and consequent frameshift into the mouse genome with the use of the CRISPR-Cas9 system (Fig 4A). A 3-bp substitution (GGG → ATT) that generated an EcoRI site for genotyping was also introduced. In silico modeling predicted almost identical secondary structures for the WT and mutant TINCR RNAs (Fig 4B), suggesting that the introduced mutations disrupt only protein function. We verified the mutations in the genome by PCR analysis combined with

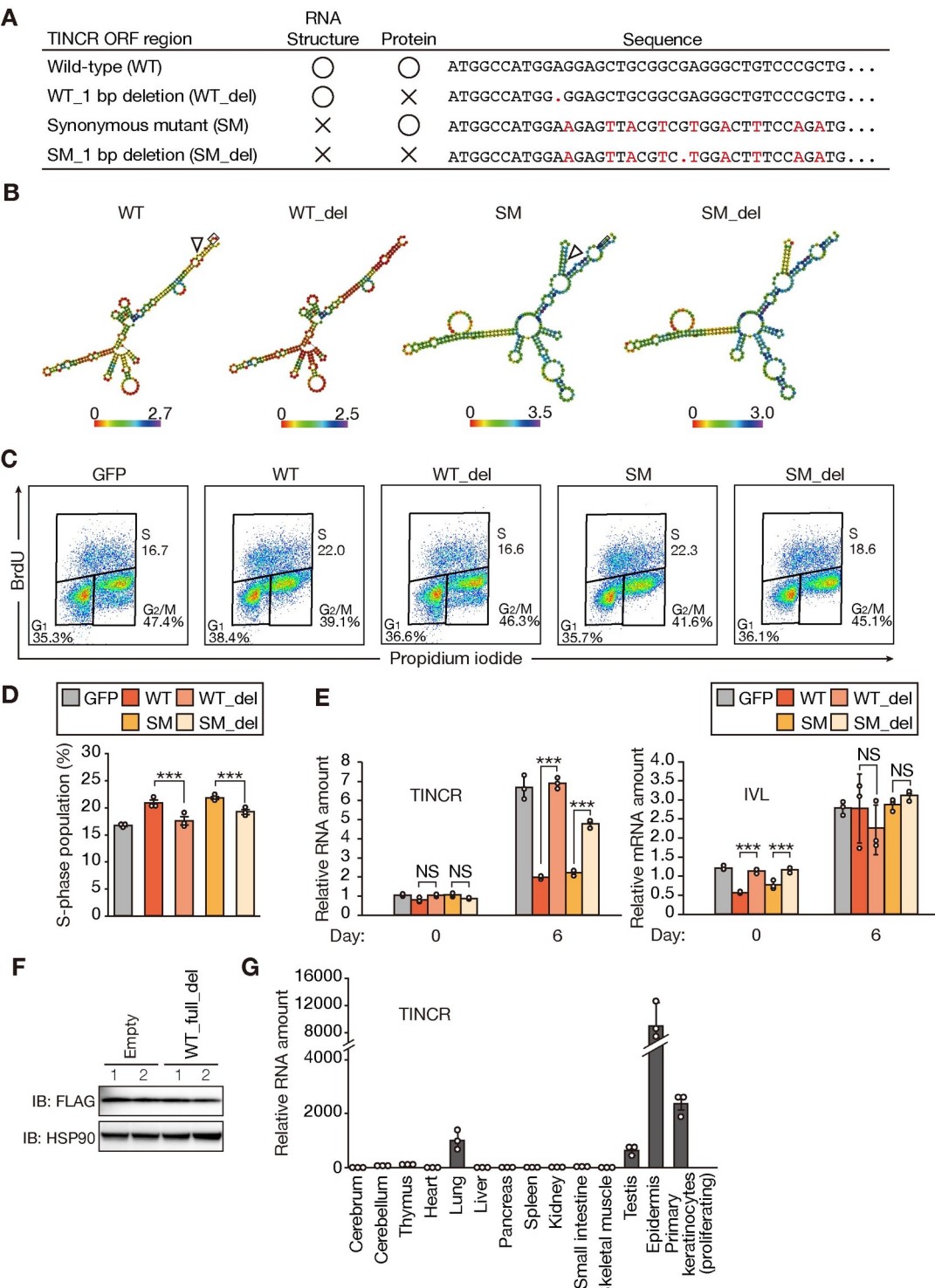

**Fig 3. TUBL expression promotes proliferation of mouse primary keratinocytes in a manner independent of the secondary structure of TINCR RNA.** (A) Summary of mutations introduced into the mouse TUBL ORF. (B) Predicted secondary structure and minimal free energy for WT, WT_del, SM, and SM_del. (C, D) Flow cytometric traces (C) and quantification (D) of BrdU incorporation for mouse primary keratinocytes stably expressing either GFP or WT, WT_del, SM, or SM_del forms of TINCR. Data in (D) are means ± SD (n = 3 independent experiments). ***p < 0.005 (Student's t test). (E) RT-qPCR analysis of TINCR and involucrin (IVL) mRNA abundance in mouse primary keratinocytes stably expressing either GFP or WT, WT_del, SM, or SM_del forms of TINCR and subjected to calcium-induced differentiation in vitro for 0 or 6 days. Data are means ± SD (n = 3

independent experiments). ***$p < 0.005$, NS (Student's t test). (F) Immunoblot analysis of HEK293T cells transiently transfected both with an expression vector for the mouse TUBL ORF with a COOH-terminal FLAG epitope tag and with either an expression vector for an almost full-length form of mouse TINCR with a 1-bp deletion in the TUBL ORF (WT_full_del) or the corresponding empty vector. Two replicates (lanes 1 and 2) are shown. (G) RT-qPCR analysis of TINCR in adult mouse tissues and actively proliferating primary mouse keratinocytes. Data are means ± SD (n = 3 independent experiments).

EcoRI digestion (Fig 4C). RT-qPCR analysis confirmed that the abundance of TINCR was not reduced in the epidermis of homozygous mutant ($Tubl^{-/-}$) mice; indeed, it was rather found to be increased ~5-fold compared with that in WT mice (Fig 4D), consistent with the proposed negative feedback effect of TUBL observed in mouse primary keratinocytes overexpressing TUBL (Fig 3E). We also confirmed the absence of TUBL in $Tubl^{-/-}$ mice with the use of MS-based multiple reaction monitoring (MRM). MRM analysis of epidermal lysates detected two different peptides derived from TUBL in $Tubl^{+/+}$ mice, but signals corresponding to such peptides were virtually absent in $Tubl^{-/-}$ mice (Figs 4E, 4F and S1). The TUBL-deficient mice of either sex manifested no obvious differences in skin morphology and histology compared with $Tubl^{+/+}$ mice at 8 weeks of age (Fig 4G). These results suggested that TUBL does not have an indispensable role in skin homeostasis under steady-state conditions.

## TUBL deficiency results in attenuated cell cycle progression in mouse keratinocytes

To examine further the role of TUBL in mouse skin, we examined gene expression profiles for the epidermis of $Tubl^{+/+}$ and $Tubl^{-/-}$ mice. RNA-seq analysis identified 26 genes whose expression was up-regulated (24) or down-regulated (2) in the epidermis of $Tubl^{-/-}$ mice compared with that of $Tubl^{+/+}$ mice on the basis of a $\log_2$[fold change] of $>0.4$ or $<-0.4$, respectively, and an adjusted p value of $<0.05$ (S2 Table). GSEA revealed that cell cycle–related gene sets were substantially down-regulated in the TUBL-deficient epidermis, whereas the expression of gene sets for keratinocyte differentiation did not differ significantly between the two genotypes (Fig 5A–5C).

We then evaluated cell cycle kinetics with a BrdU incorporation assay in primary keratinocytes established from the two groups of mice. The proportion of cells in S phase of the cell cycle was markedly decreased for the TUBL-deficient primary keratinocytes compared with WT control cells (Fig 5D). Examination of the expression of differentiation marker genes by RT-qPCR analysis revealed that the differentiation status of the epidermis was similar for $Tubl^{+/+}$ and $Tubl^{-/-}$ mice (Fig 5E). Furthermore, an in vitro differentiation assay with primary keratinocytes showed no significant difference in expression of the involucrin gene between the two genotypes (Fig 5F). Together, these results suggested that the loss of TUBL results in attenuated progression of the cell cycle in mouse keratinocytes.

## Loss of TUBL results in delayed recovery from skin injury

Delayed cell cycle progression of keratinocytes can affect wound healing after injury in mice [6–8]. We therefore performed biopsy punches on the skin of TUBL-deficient mice in order to evaluate their ability to recover. $Tubl^{-/-}$ mice manifested delayed wound closure from days 4 to 10 postinjury compared with WT control animals, although the extent of wound closure did not differ significantly between the two genotypes thereafter (Fig 6). These results are consistent with the attenuated proliferation of TUBL-deficient keratinocytes (Fig 5D) as well as with the fact that re-epithelialization, during which keratinocytes proliferate to cover a wound, generally occurs 3 to 10 days after injury in mice [6]. Our findings thus suggest that TUBL

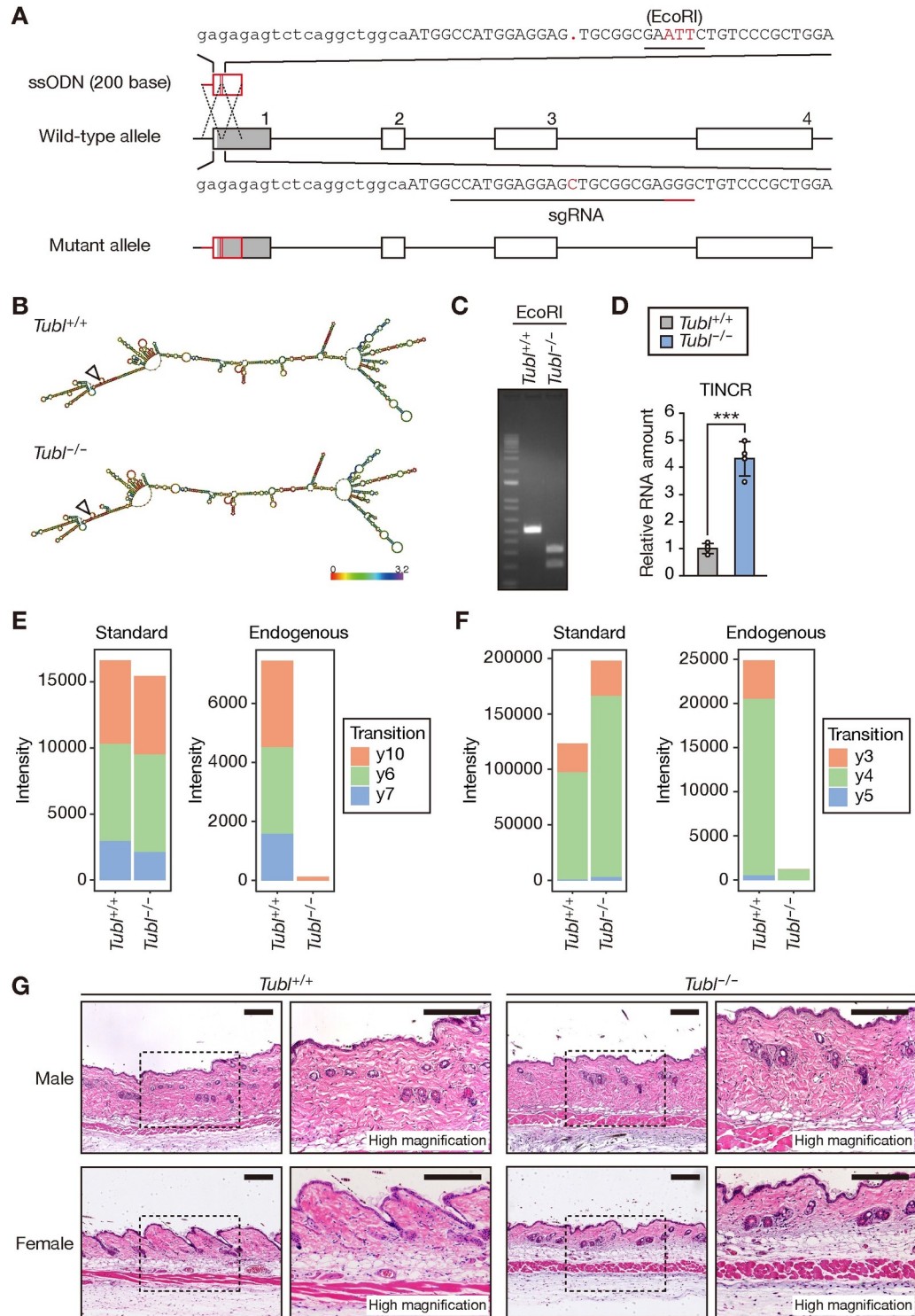

**Fig 4. Generation of TUBL-deficient mice.** (A) Schematic representation of the WT *TINCR* allele, the single-stranded oligodeoxynucleotide (ssODN), and the mutant allele after homologous recombination. Exons are denoted by numbered boxes. The single guide RNA (sgRNA) for the CRISPR-Cas9 system and its protospacer adjacent motif (PAM) are indicated by contiguous black and red underlines, respectively. The TUBL ORF is represented by the gray shading in the box corresponding to exon 1 of *TINCR*. (B) Predicted secondary structure and minimal free energy for WT TINCR and the mutant form generated by the CRISPR-Cas9 system for establishment of *Tubl⁻/⁻* mice. The triangle indicates the 5' end of the transcript. (C) PCR analysis of genomic DNA from the tail of mice of the indicated genotypes. The PCR

products were digested with EcoRI before electrophoresis. (D) RT-qPCR analysis of TINCR in the epidermis of $Tubl^{+/+}$ and $Tubl^{-/-}$ mice. Data are means ± SD (n = 3 independent experiments). $^{***}$p < 0.005 (Student's t test). (E, F) Signal intensity of extracted ion chromatograms for mouse TUBL peptides in MRM analysis. The analysis detected two different peptides derived from mouse TUBL with the amino acid sequences AQLVGQGVSSWR (E) and DTLSDLR (F). "Standard" indicates an internal standard corresponding to stable isotope–labeled recombinant mouse TUBL. "Endogenous" indicates the endogenous TUBL peptides in epidermal lysates prepared from $Tubl^{+/+}$ or $Tubl^{-/-}$ mice. Tryptic peptides derived from endogenous TUBL were mixed with tryptic peptides derived from the isotopically labeled recombinant protein and were then applied to MS analysis. Extracted ion chromatograms for each transition are shown in S1 Fig. (G) Hematoxylin-eosin staining of the skin of $Tubl^{+/+}$ or $Tubl^{-/-}$ mice at 8 weeks of age. The boxed regions in the left panels of each set are shown at higher magnification in the right panels. Scale bars, 300 μm.

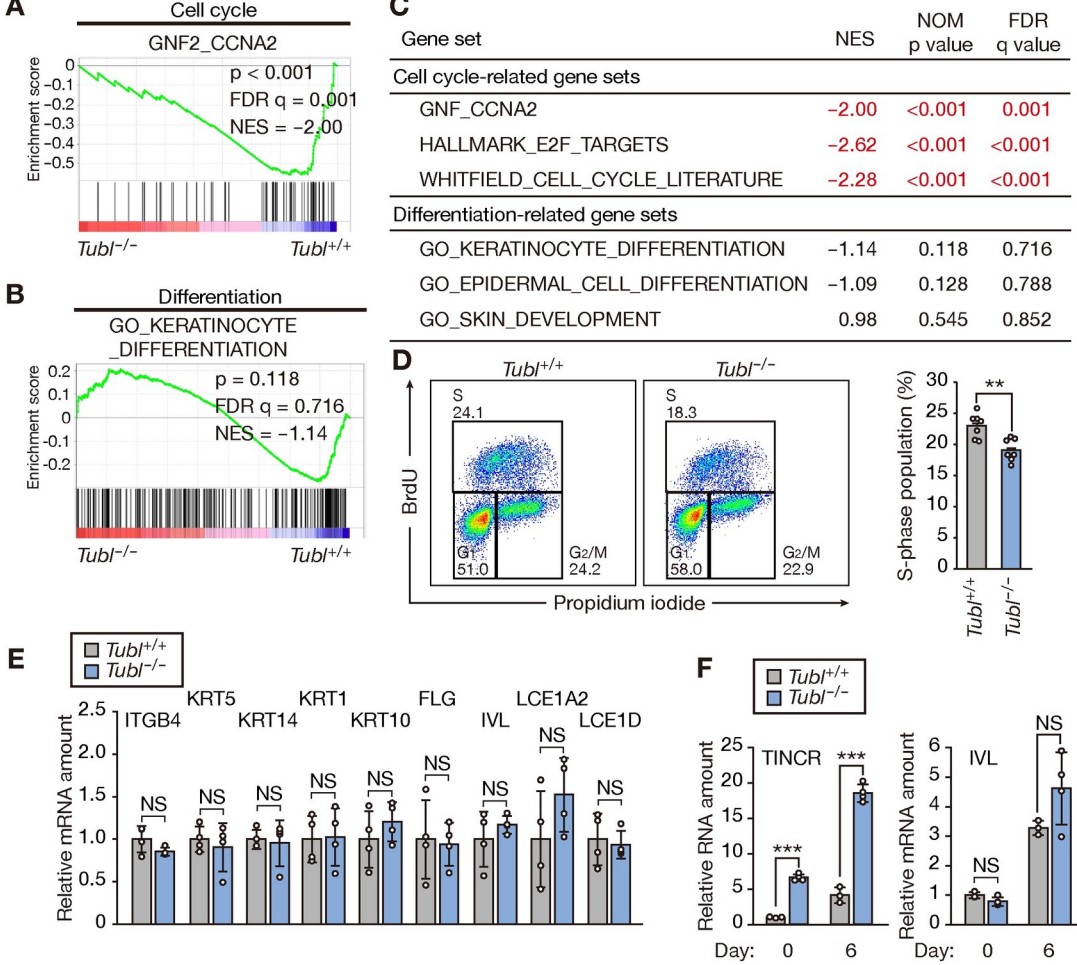

**Fig 5. TUBL deficiency delays cell cycle progression in mouse primary keratinocytes.** (A, B) GSEA plots for gene sets related to the cell cycle (A) or keratinocyte differentiation (B) constructed from RNA-seq data for the epidermis of $Tubl^{+/+}$ or $Tubl^{-/-}$ mice at 8 weeks of age. FDR, false discovery rate; NES, normalized enrichment score. (C) Results of GSEA for gene sets related to the cell cycle or keratinocyte differentiation. NOM, nominal. (D) Flow cytometric traces and quantification of BrdU incorporation in primary keratinocytes from $Tubl^{+/+}$ or $Tubl^{-/-}$ mice. Data in the right panel are means ± SD (n = 7 or 8 independent experiments). $^{**}$p < 0.01 (Student's t test). (E) RT-qPCR analysis of keratinocyte differentiation–related gene expression in the epidermis of $Tubl^{+/+}$ and $Tubl^{-/-}$ mice. Data are means ± SD (n = 4 independent experiments). NS, Student's t test. (F) RT-qPCR analysis of TINCR and involucrin mRNA abundance in mouse primary keratinocytes established from $Tubl^{+/+}$ or $Tubl^{-/-}$ mice and subjected to calcium-induced differentiation in vitro for 0 or 6 days. Data are means ± SD (n = 3 or 4 independent experiments). $^{***}$p < 0.005, NS (Student's t test).

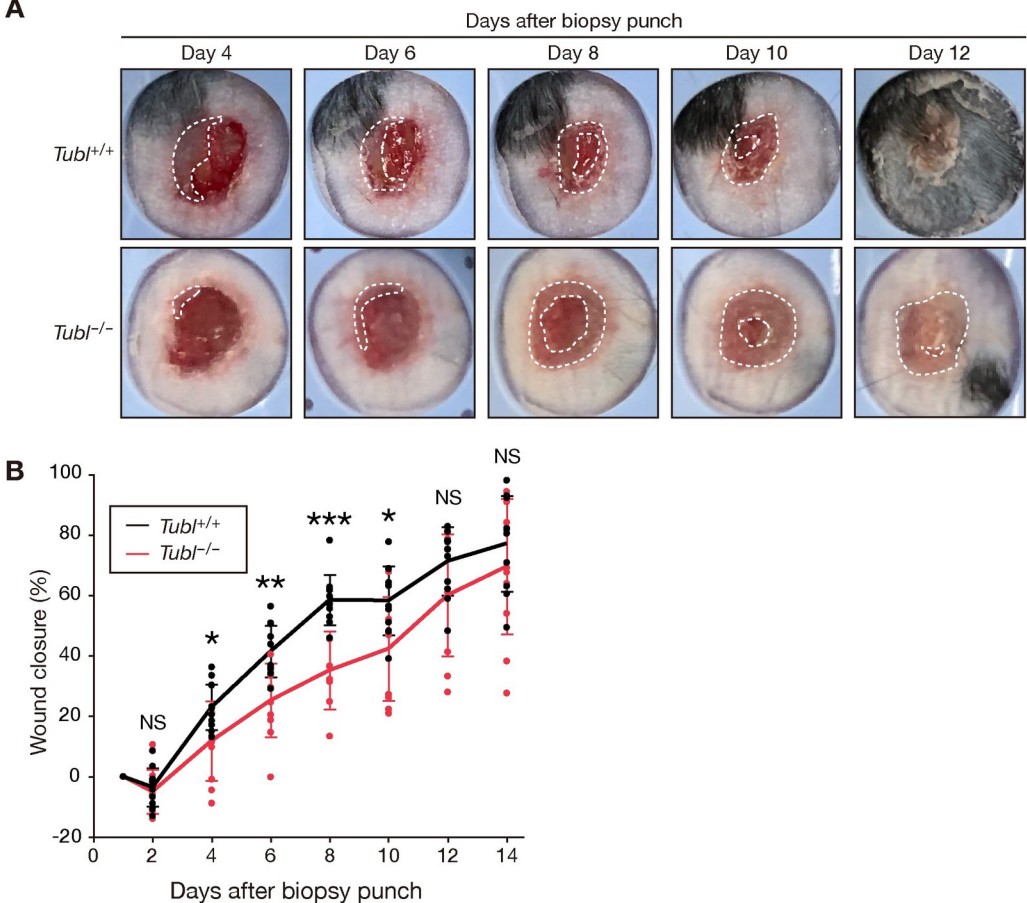

**Fig 6. Loss of TUBL results in delayed recovery after skin injury.** (A) Representative macroscopic views of cutaneous ulcers of $Tubl^{+/+}$ and $Tubl^{-/-}$ mice at 4, 6, 8, 10, and 12 days after wounding with a biopsy punch. The area demarcated by white dashed lines indicates the area of re-epithelialization. (B) Time course of wound healing in $Tubl^{+/+}$ (n = 11) and $Tubl^{-/-}$ (n = 10) mice. Data are means ± SD. $^{*}$p < 0.05, $^{**}$p < 0.01, $^{***}$p < 0.005, NS versus corresponding value for $Tubl^{-/-}$ mice (Student's t test).

contributes to the maintenance of skin homeostasis by promoting the proliferation of keratinocytes after injury.

## TUBL binds to the proteasome complex

To explore the molecular function of TUBL, we purified protein complexes containing TUBL from NHEKs or HaCaT (human keratinocyte) cells transiently expressing human TUBL with a COOH-terminal FLAG tag. Immunoprecipitates prepared from the cells with antibodies to FLAG were subjected to semiquantitative liquid chromatography and tandem MS (LC-MS/MS) analysis. We identified several proteins that were specifically associated with TUBL in that they were detected specifically in at least one replicate of the TUBL-(C)FLAG samples or the average iBAQ (intensity-based absolute quantification) value for the three TUBL-(C)FLAG replicates was >2-fold as great as that for the control samples (Fig 7A and S3 Table). Gene ontology (GO) term analysis of the TUBL-associated proteins revealed enrichment of several cell cycle–related terms, such as those involving "ubiquitin-protein ligase activity involved in mitotic cell cycle" (Fig 7B). However, most genes constituting such GO terms actually encode

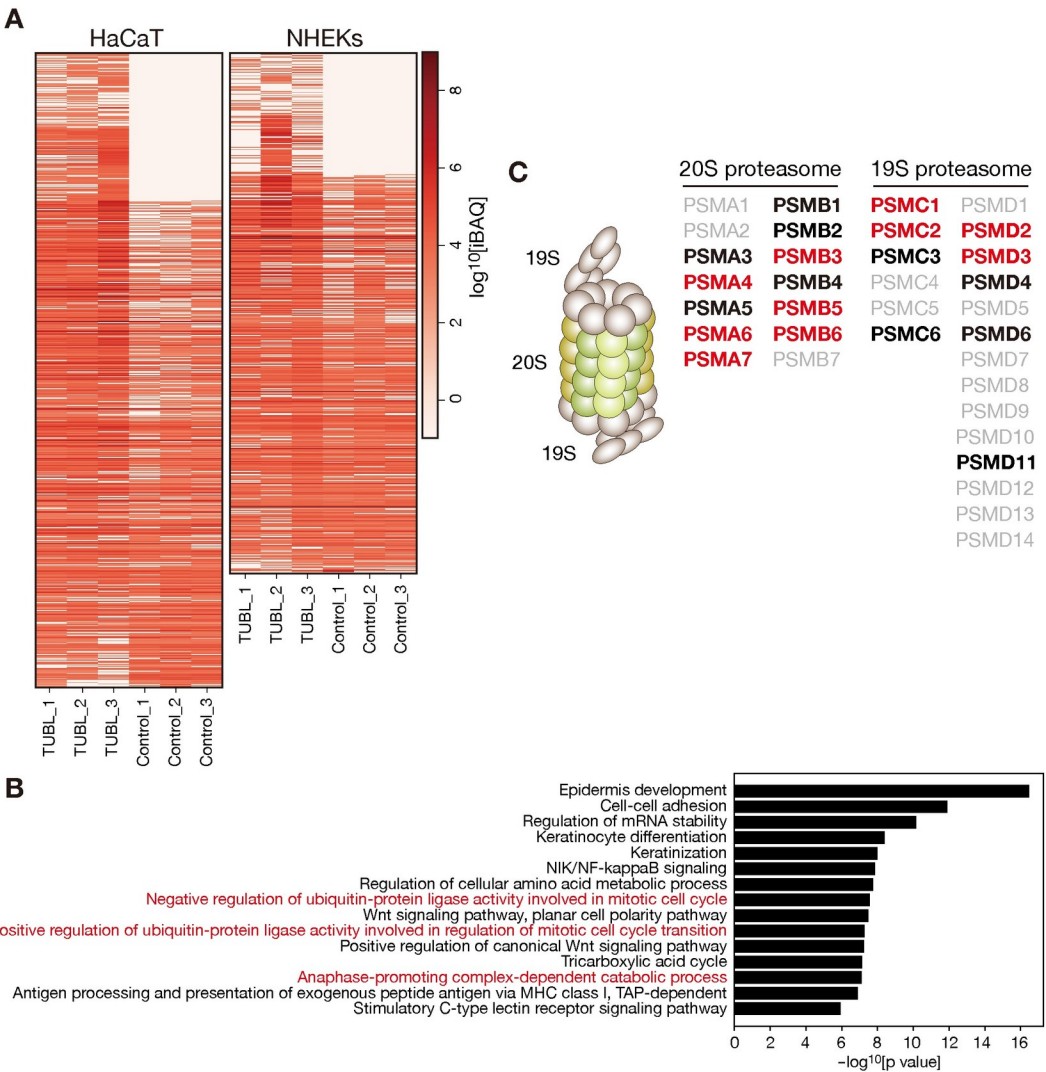

**Fig 7. Identification of TUBL binding proteins.** (A) Heat map of iBAQ (intensity-based absolute quantification) values for HaCaT cells or NHEKs transiently expressing human TUBL with a COOH-terminal FLAG tag or transfected with the corresponding empty vector (Control). Immunoprecipitates prepared from the cells with antibodies to FLAG were subjected to semiquantitative LC-MS/MS analysis. Data are shown for three biological replicates. (B) GO analysis of identified TUBL binding proteins. Representative findings are presented. (C) Schematic representation of the proteasome complex (left) and lists of proteasome subunits identified as TUBL binding proteins (right). 19S and 20S indicate the 19S regulatory particle and 20S catalytic particle of the proteasome, respectively. Red and black characters indicate subunits identified in both HaCaT cells and NHEKs or in one of the two cell types, respectively.

subunits of the proteasome complex and were commonly identified in both NHEKs and HaCaT cells (Fig 7C and S4 Table).

TUBL is a type 2 Ubl protein but does not harbor any other apparent domains. It is therefore a "Ubl-only" protein that associates with the proteasome. In contrast, RAD23 and DSK2, both of which are representative type 2 Ubl proteins, contain a Ubl domain that associates with the proteasome as well as a Uba (ubiquitin-associated) domain that interacts with ubiquitylated proteins, and they therefore facilitate the recruitment of ubiquitylated proteins to the proteasome for efficient proteolysis [32]. Given its lack of a Uba domain, TUBL is expected to lack such an ability, suggesting that it might function as a negative regulator that modifies the

interaction between authentic Ubl-Uba proteins and the proteasome. Elucidation of the detailed molecular mechanism by which TUBL regulates cell cycle progression awaits further study, however.

## Discussion

TINCR, also known as placenta-specific 2 (non–protein coding), has been recognized as a lncRNA, and it was indeed shown to function as such [33]. On the other hand, an ORF of TINCR was registered as a predicted protein in the UniProt database (accession number: A0A1B0GVN0). Furthermore, peptide fragments corresponding to the predicted ORF were identified in the cornified envelope of human skin by proteomics analysis in the DDA mode [29]. The roles of TINCR and the expression of its putative translational product have thus remained unclear.

We have now shown that the ORF of TINCR encodes a novel ubiquitin-like protein, which we designated TUBL. Insertion of an epitope tag sequence in the endogenous ORF as well as targeted proteomics analysis demonstrated the expression of this protein. Forced expression of TUBL in NHEKs and mouse primary keratinocytes promoted cell cycle progression in a manner independent of the RNA secondary structure of TINCR. Moreover, TUBL-deficient mice manifested a significant delay in wound healing after injury with a biopsy punch, with this delay being associated with attenuation of keratinocyte proliferation.

The barrier function of skin is conferred by terminally differentiating keratinocytes. This differentiation process is known as cornification, results in formation of the cornified envelope, and occurs in the suprabasal layer of the epidermis. The cornified envelope consists of proteins that are highly cross-linked by transglutaminase and which are surrounded by specific lipids, and it provides a mechanical and permeability barrier [34]. Whereas TUBL was detected only in the insoluble cornified envelope, not in the soluble fraction of the epidermis, in the study by Eckhart et al. [29], we detected endogenous TUBL expression in the soluble fraction by both immunoblot and targeted proteomics analyses. Given the apparent high level of expression of TUBL in both proliferating and differentiated keratinocytes, the protein may contribute to cell cycle regulation in the cytoplasm of proliferating keratinocytes, whereas, in terminally differentiated keratinocytes, it might have other functions such as formation of the cornified envelope.

The abundance of TINCR is up-regulated in various cancers, in which it promotes proliferation of the cancer cells [10–21]. In the present study, we found that forced expression of TUBL promoted keratinocyte proliferation in a manner independent of the RNA secondary structure of TINCR, suggesting that promotion of cancer cell proliferation by TINCR is also likely mediated by TUBL protein rather than by TINCR RNA. It will thus be of interest to generate antibodies specific for TUBL so as to investigate potential changes in the expression of TUBL in human cancers. The development of inhibitors of TUBL might prove beneficial for cancer treatment, whereas TUBL supplementation may be effective for promotion of skin wound healing. The risks and benefits of such potential therapeutic approaches await further investigation.

## Methods

### Ethical statement

All mouse experiments were approved by the Animal Ethics Committee of Kyushu University (Accession number A20-169).

## Animals

For generation of TUBL knockout mice and knock-in mice with a FLAG tag sequence at the COOH-terminus of TUBL, ribonucleoprotein (RNP) was prepared by mixing the CRISPR RNA (crRNA) and trans-activating crRNA (tracrRNA) with recombinant Cas9 protein (Integrated DNA Technologies) (S5 Table). Single-stranded oligodeoxynucleotide (ssODN) homology repair templates were synthesized as 200-nt sequences (S5 Table). Mouse zygotes were subjected to electroporation with the RNP and ssODN.

## Cell culture, in vitro differentiation, and transfection

HEK293T and HeLa cells were obtained from American Type Culture Collection and were checked for mycoplasma contamination with the use of MycoAlert (Lonza). They were cultured under an atmosphere of 5% $CO_2$ at 37˚C in Dulbecco's modified Eagle's medium supplemented with 10% fetal bovine serum (Life Technologies) and antibiotics. For transient expression of human or mouse TUBL and TINCR, human NEDD8, or mutants of these proteins, cDNAs were subcloned into pcDNA3 (Invitrogen) with the use of a NEBuilder HiFi DNA Assembly Master Mix kit (New England BioLabs) and transfection was performed with the use of the X-tremeGENE 9 DNA Transfection Reagent (Sigma-Aldrich). HaCaT cells were obtained from COSMO BIO and cultured in Dulbecco's modified Eagle's medium (calcium free) supplemented with 10% fetal bovine serum (Life Technologies) and antibiotics. NHEKs were obtained from Lonza, and early-passage cells (fewer than seven passages from initial plating) were maintained with the use of a KGM Gold Keratinocyte Growth Medium BulletKit (Lonza). For the in vitro differentiation assay, NHEKs at 100% confluence were exposed to 1.3 mM $CaCl_2$ for 6 days. For siRNA transfection, NHEKs were incubated for 24 h in six-well plates with 5 nM siRNA (Lincode nontargeting pool, Lincode human TINCR [257000] siRNA-SMART, Silencer Select Negative Control No.1 siRNA, Silencer Select Pre-Designed siRNA human TINCR [s48832], or Silencer Select Pre-Designed siRNA human TINCR [s195902]) and with the Lipofectamine RNAiMax reagent (Thermo Fisher Scientific). The cells were subjected to the in vitro differentiation assay.

## RT-qPCR analysis

The epidermis was isolated by incubation of skin for 30 min at 37˚C in 3.8% ammonium thiocyanate. Total RNA was extracted from the epidermis or cells with the use of Isogen (Nippon Gene), purified with the use of a PureLink RNA Mini Kit (Thermo Fisher Scientific), and subjected to RT with the use of ReverTra Ace qPCR RT Master Mix with gDNA Remover (Toyobo). The resulting cDNA was subjected to real-time PCR analysis with Luna Universal qPCR Master Mix (New England BioLabs) and specific primers in a StepOnePlus Real-Time PCR System (Applied Biosystems). The sequences of PCR primers are provided in S5 Table. The amounts of TINCR or of target mRNAs were normalized by that of glyceraldehyde-3-phosphate dehydrogenase (GAPDH) mRNA.

## Immunoblot analysis

Protein samples were subjected to SDS–polyacrylamide gel electrophoresis (PAGE) on 5–20% ExtraPAGE One Precast Gels (Nacalai Tesque). Membranes were incubated consecutively with primary antibodies and horseradish peroxidase–conjugated secondary antibodies (Promega), and signals were visualized with SuperSignal West Pico PLUS or Dura (Thermo Fisher Scientific) reagents and a ChemiDoc imaging system (Bio-Rad). Primary antibodies included

the following: V5 tag (Thermo Fisher Scientific), FLAG tag (Merck), HSP70 (BD Biosciences), and HSP90 (BD Biosciences).

## Immunoprecipitation assay

The epidermis was isolated by incubation of skin for 16 h at 4˚C in keratinocyte basal medium (EpiLife CF Kit, EpiLife Defined Growth Supplement [EDGS]) containing dispase (4 mg/ml) (Thermo Fisher Scientific). Total protein was extracted from the epidermis and other tissues with the use of a lysis buffer (50 mM Tris-HCl [pH 7.5], 150 mM NaCl, 1% Triton X-100, 1× proteinase inhibitor cocktail (Roche)) and a Biomasher II (BMBio), followed by centrifugation at $20,000 \times g$ for 20 min at 4˚C. The resulting supernatant was incubated with rotation for 1 h at 4˚C with agarose bead–conjugated antibodies to FLAG (Merck). The beads were washed three times with a wash buffer (50 mM Tris-HCl [pH 7.5], 150 mM NaCl, 0.1% Triton X-100, 1× proteinase inhibitor cocktail), and the bound proteins were eluted from the beads with wash buffer containing FLAG peptide (0.5 mg/ml) (Merck) and then subjected to immunoblot analysis as described above.

## Immunofluorescence staining

Cells grown on coverslips were fixed with 4% paraformaldehyde in phosphate-buffered saline (PBS) for 10 min at room temperature, washed twice with PBS, and incubated for 16 h at 4˚C with antibodies to FLAG (Merck) diluted in PBS containing 0.5% Triton X-100. They were then washed three times with PBS, incubated for 1 h at room temperature in the dark with Alexa Fluor 488–conjugated secondary antibodies (Thermo Fisher Scientific) diluted in PBS containing 0.5% Triton X-100, and then incubated for 5 min at room temperature in the dark with PBS containing DAPI. After two washes with PBS, the coverslips were mounted in Fluoromount and examined with an LSM 700 fluorescence microscope (Zeiss).

## Histopathology

Tissue was fixed with 4% paraformaldehyde in PBS, embedded in paraffin, sectioned with a cryostat at a thickness of 3 μm, and stained with hematoxylin-eosin. Sections were examined with a differential interference contrast microscope.

## Isolation and culture of primary mouse keratinocytes

Primary keratinocytes were isolated from the tail of 8-week-old mice. Isolation and culture of the cells were performed as described previously [35]. The cells were maintained in keratinocyte basal medium (EpiLife CF Kit, EpiLife Defined Growth Supplement [EDGS]) and were plated in dishes coated with the use of a Coating Matrix Kit (Thermo Fisher Scientific) to promote adherence. For the in vitro differentiation assay, keratinocytes at 100% confluence were exposed for 6 days to 1.3 mM $CaCl_2$ in keratinocyte basal medium without EDGS.

## Lentivirus expression system

GFP or human TUBL cDNA was subcloned into pLVSIN/puro, and the resulting constructs together with pMD2.G, pMDLg/pRRE, and pRSV-Rev (Addgene) were introduced into HEK293T cells with the use of the X-tremeGENE 9 reagent (Roche). Culture supernatants containing recombinant lentiviruses were harvested and used to infect NHEKs for 24 h in the presence of polybrene (2 μg/ml). The cells were then subjected to selection by incubation for an additional 24 h in medium containing puromycin (2.5 μg/ml), cultured without puromycin for 48 h, and then applied to assays.

### Retrovirus expression system

GFP or WT or mutant mouse TUBL cDNAs were subcloned into pCX4/puro, and the resulting constructs were introduced into Plat E packaging cells with the use of the FuGENE6 reagent (Roche). Culture supernatants containing recombinant ecotropic retroviruses were harvested and used to infect mouse primary keratinocytes by incubation for 24 h in the presence of polybrene (2 μg/ml). The cells were cultured for an additional 24 h in virus-free medium, subjected to selection for 24 h in medium containing puromycin (2.5 μg/ml), and incubated for 72 h in puromycin-free medium before application to assays.

### RNA-seq analysis

The epidermis was collected by incubation of skin in 3.8% ammonium thiocyanate. Total RNA was extracted from the epidermis or from NHEKs stably expressing GFP or TUBL and was purified with the use of Isogen and a PureLink RNA Mini Kit. The quality of the purified RNA was monitored with a 2100 Bioanalyzer (Agilent), mRNA was selected with the use of a NEBNext Poly(A) mRNA Magnetic Isolation Module (New England BioLabs), and libraries were prepared with the use of a NEBNext Ultra Directional RNA Library Prep Kit for Illumina (New England BioLabs). The cDNAs were sequenced with a HiSeq 1500 system (Illumina). Single-end reads were mapped against the mouse (mm10) or human (hg38) genome with the use of HISAT2, and data normalization was performed with DESeq2. The expression data were analyzed with GSEA v4.1.0 software. The gene sets for GSEA were obtained from the Molecular Signatures Database v4.0 distributed at the GSEA Web site.

### Immunoaffinity purification and LC-MS/MS analysis

HaCaT cells and NHEKs were collected at 48 h after transfection with the use of the FuGENE HD reagent (Promega). Cell lysates were subjected to immunoprecipitation with antibodies to FLAG and the precipitated proteins were isolated by elution as described above. The purified proteins were fractionated by SDS-PAGE on a 10% gel and stained with silver. Protein bands were excised from the gel and subjected to in-gel digestion with trypsin, and the resulting peptides were dissolved in a solution comprising 0.1% trifluoroacetic acid and 2% acetonitrile for analysis with an LTQ Orbitrap Velos Pro mass spectrometer (Thermo Fisher Scientific) coupled with a nanoLC instrument (Advance, Michrom BioResources) and HTC-PAL autosampler (CTC Analytics). Acquired MS data were analyzed with MaxQuant and the human UniProt KB database. GO analysis was performed with the use of DAVID [36].

### MRM analysis

To identify target peptides for MRM analysis, mouse TUBL with a COOH-terminal FLAG tag was immunoprecipitated from transfected HEK293T cells with M2-conjugated agarose beads (Sigma-Aldrich) and then eluted from the beads with the FLAG peptide (0.5 mg/ml). The eluted material was fractionated by SDS-PAGE on a 13.5% gel, and gel slices corresponding to a molecular size of 10 to 15 kDa were subjected to reduction and alkylation followed by digestion with trypsin. The resultant peptides were subjected to LC-MS/MS analysis in the DDA mode with a TripleTOF5600 instrument. Peak lists (mgf) generated by the AB SCIEX MS Data Converter were used to search a database, mouse UniProt containing a mouse TUBL protein sequence, with the use of the Mascot algorithm (Matrix Science). The search was conducted with the following parameter settings: trypsin as enzyme used, an allowed number of missed cleavages of two, and carbamidomethylation of Cys as a fixed modification. Oxidized methionine was searched as a variable modification. Precursor mass tolerance was 50 ppm, and

tolerance of MS/MS ions was 0.02. We imported all significant hits into an in-house relational database written in MySQL. MS/MS spectra assigned to peptides with more than six amino acids were selected from the database and converted into MRM transitions and a spectral library for iMPAQT-Quant [37]. Hexahistidine-tagged mouse TUBL was then expressed in an *Escherichia coli* expression strain that was auxotrophic for arginine and lysine and grown in the presence of $^{13}C_6/^{15}N_4$-Arg and $^{13}C_6/^{15}N_2$-Lys [38], and the recombinant protein was purified with the use of Ni-resin (Probond, Invitrogen). The epidermis was isolated by incubation of mouse skin for 16 h at 4˚C in keratinocyte basal medium containing dispase (4 mg/ml), extracts prepared with lysis buffer were fractionated by SDS-PAGE on a 13.5% gel, and gel slices corresponding to a molecular size of <13 kDa were subjected to reduction and alkylation followed by digestion with trypsin. The resulting peptides were mixed with tryptic peptides derived from the isotopically labeled recombinant protein and stored until MS analysis. MRM analysis was performed with a QTRAP6500 system (SCIEX) operated in the positive-ion mode. Typical parameters were set as follows: spray voltage of 2300 V, curtain gas setting of 10, collision gas setting of high, ion source gas 1 setting of 30, and interface heater temperature of 160˚C. The collision energy (CE) was calculated with the following formulae: CE = (0.044 × $m/z1$) + 5.5 and CE = (0.051 × $m/z1$) + 0.5, where $m/z1$ is the mass/charge ($m/z$) ratio of the precursor ion, for doubly and triply charged precursor ions, respectively. The collision cell exit potential (CXP) was calculated according to the following formula: CXP = (0.0391 × $m/z2$)– 2.2334, where $m/z2$ is the $m/z$ of the fragment ion. The declustering potential (DP) was set to 50, and the entrance potential (EP) was set to 10. Resolution for Q1 and Q3 was set to "unit" (half-maximal peak width of 0.7 Da). The scheduled MRM option was used for all data acquisition with a target scan time of 6.7 s and MRM detection windows of 300 s for verification of MRM assays. Raw data were analyzed by iMPAQT-Quant with the corresponding spectral library. Peak groups were scored on the basis of cosine similarity with the MS/MS spectra obtained in DDA, a peak coelution of at least three fragment ions for each peptide, the presence or absence of interfering ions, and intensity. Finally, all traces were manually checked to eliminate inadequate transitions.

## Cell cycle analysis

Cells were pulsed with BrdU (10 μg/ml) (Sigma) for 30 min, stained with Alexa Fluor 488– or allophycocyanin-labeled antibodies to BrdU (BioLegend) and with propidium iodide (Nacalai), and then analyzed with a FACSVerse flow cytometer (BD).

## Prediction of RNA secondary structure

The Vienna RNA package was applied to calculate structures of minimum free energy for each RNA [39]. The sequences cloned into the pCX4 retrovirus expression vector were 5′- ATGGC CATGGAGGAGCTGCGGCGAGGGCTGTCCCGCTGGAAGCGCTACCACATCAAGGT ACACCTAGCCGACGAGGCATTGCTACTGCCACTGACCGTGCGGCCCCGAGACACA CTGAGTGACCTGCGCGCACAGCTAGTGGGCCAGGGTGTAAGCTCTTGGCGCAGA ACCTTCTACTACAACTCTAGGCCTCTGCCTGACCATCAGACAGTTCGCGAAGCT CGCCTGCAGGACGGCTCAGTATTACTTCTGCTCAGCGACACCAGGTAG-3′ for wild-type (WT) TUBL, 5′- ATGGCCATGGGGGAGCTGCGGCGAGGGCTGTCCCGCTGGAA GCGCTACCACATCAAGGTACACCTAGCCGACGAGGCATTGCTACTGCCACTGAC CGTGCGGCCCCGAGACACACTGAGTGACCTGCGCGCACAGCTAGTGGGCCAGG GTGTAAGCTCTTGGCGCAGAACCTTCTACTACAACTCTAGGCCTCTGCCTGACCAT CAGACAGTTCGCGAAGCTCGCCTGCAGGACGGCTCAGTATTACTTCTGCTCAGCGA CACCAGGTAG-3′ for WT TUBL containing a 1-bp deletion (WT_del), 5′- ATGGCCATGG

AAGAGTTACGTCGTGGACTTTCCAGATGGAAACGATATCATATAAAAGTACATTT
GGCTGATGAAGCTTTACTATTACCACTTACGGTTAGACCCAGAGATACATTGTCTG
ATCTCCGTGCTCAACTTGTCGGCCAAGGTGTCTCCTCTTGGCGCAGAACCTTTTAT
TATAATTCTAGACCTCTCCCTGATCATCAAACGGTTCGAGAAGCCAGGCTTCAAG
ATGGTTCCGTACTCTTACTCCTCTCAGATACCCGTTGA-3′ for the TUBL synonymous
mutant (SM), and 5′- ATGGCCATGGAAGAGTTACGTCTGGACTTTCCAGATGGAAAC
GATATCATATAAAAGTACATTTGGCTGATGAAGCTTTACTATTACCACTTACGGTT
AGACCCAGAGATACATTGTCTGATCTCCGTGCTCAACTTGTCGGCCAAGGTGTCT
CCTCTTGGCGCAGAACCTTTTATTATAATTCTAGACCTCTCCCTGATCATCAAACG
GTTCGAGAAGCCAGGCTTCAAGATGGTTCCGTACTCTTACTCCTCTCAGATACCC
GTTGA-3′ for the TUBL synonymous mutant containing a 1-bp deletion (SM_del). Both WT
and SM sequences encode the same protein product: MAMEELRRGLSRWKRYHIKVHLAD
EALLLPLTVRPRDTLSDLRAQLVGQGVSSWRRTFYYNSRPLPDHQTVREARLQDGSVL
LLLSDTR.

## Wound healing assay

A 6-mm punch biopsy tool was used to inflict full-thickness skin wounds on the back of mice
that extended through the panniculus carnosus under inhalation anesthesia with isoflurane. A
circular silicone splint (inner diameter of 10 mm, outer diameter of 16 mm) was placed on
each wound, and adhesive (Aron Alpha high speed EX, Konishi) was then applied to maintain
the position of the splint. The use of the splint minimized wound contracture and allowed for
healing through re-epithelialization and formation of granulation tissue.

## Statistical analysis

Quantitative data are presented as means ± SD as indicated and were compared between
groups with the two-tailed Student's t test as performed with Microsoft Excel or JMP Pro soft-
ware. A p value of <0.05 was considered statistically significant. All numerical data underlying
graphs or statistics are provided in S6 Table.

## Supporting information

**S1 Fig. Detection of TUBL protein expression in epidermal lysates by MS-based MRM
analysis.** Extracted ion chromatograms are shown for TUBL-derived peptides in MRM analy-
sis. MRM analysis detected two different TUBL-derived peptides with the amino acid
sequences AQLVGQGVSSWR (A) and DTLSDLR (B). "Standard" indicates signals for the
internal standard, a stable isotope–labeled recombinant TUBL protein expressed in and puri-
fied from bacteria. "Endogenous" indicates signals for peptides derived from endogenous
TUBL in epidermal lysates prepared from $Tubl^{+/+}$ or $Tubl^{-/-}$ mice. The purple vertical lines in
the Endogenous chromatograms indicate the retention time of each endogenous TUBL pep-
tide.
(EPS)

**S1 Table. Differentially expressed genes, including up-regulated and down-regulated
genes, in NHEKs infected with a lentivirus encoding TUBL compared with those infected
with a lentivirus encoding GFP.**
(XLSX)

**S2 Table. Differentially expressed genes, including up-regulated and down-regulated
genes, in epidermis of $Tubl^{-/-}$ mice compared with that of $Tubl^{+/+}$ mice.**
(XLSX)

**S3 Table. iBAQ values for TUBL binding proteins identified in HaCaT cells and NHEKs.**
(XLSX)

**S4 Table. Results for GO term analysis of identified TUBL binding proteins.**
(XLSX)

**S5 Table. Oligonucleotides used in the present study.**
(XLSX)

**S6 Table. All numerical data underlying graphs or statistics.**
(XLSX)

## Acknowledgments

We thank the members of the Department of Molecular and Cellular Biology, Medical Institute of Bioregulation, Kyushu University, and the Research Promotion Unit, Medical Institute of Bioregulation, Kyushu University for technical assistance as well as A. Ohta for help with preparation of the manuscript. Computations were performed in part on the NIG supercomputer at ROIS National Institute of Genetics.

## Author Contributions

**Conceptualization:** Akihiro Nita, Akinobu Matsumoto, Kenji Kabashima, Atsushi Hatano, Masaki Matsumoto, Keiichi I. Nakayama.

**Data curation:** Akihiro Nita, Akinobu Matsumoto, Kazuya Ichihara, Daisuke Saito, Mikita Suyama, Tomoharu Yasuda, Bumpei Katayama, Toshiyuki Ozawa, Teruasa Murata, Teruki Dainichi, Atsushi Hatano, Masaki Matsumoto.

**Formal analysis:** Akihiro Nita, Akinobu Matsumoto, Ronghao Tang, Chisa Shiraishi, Kazuya Ichihara, Daisuke Saito, Tomoharu Yasuda, Bumpei Katayama, Toshiyuki Ozawa, Teruasa Murata, Teruki Dainichi, Atsushi Hatano, Masaki Matsumoto.

**Funding acquisition:** Akinobu Matsumoto, Keiichi I. Nakayama.

**Investigation:** Akihiro Nita, Akinobu Matsumoto, Ronghao Tang, Chisa Shiraishi, Kazuya Ichihara, Tomoharu Yasuda, Bumpei Katayama, Toshiyuki Ozawa, Teruasa Murata, Teruki Dainichi, Atsushi Hatano, Masaki Matsumoto.

**Methodology:** Akihiro Nita, Akinobu Matsumoto, Kazuya Ichihara, Daisuke Saito, Bumpei Katayama, Toshiyuki Ozawa, Teruasa Murata, Teruki Dainichi, Atsushi Hatano, Masaki Matsumoto.

**Project administration:** Akinobu Matsumoto, Kenji Kabashima, Keiichi I. Nakayama.

**Resources:** Akihiro Nita, Akinobu Matsumoto, Gaku Tsuji, Masutaka Furue, Keiichi I. Nakayama.

**Software:** Kazuya Ichihara, Daisuke Saito, Mikita Suyama, Atsushi Hatano, Masaki Matsumoto.

**Supervision:** Akinobu Matsumoto, Gaku Tsuji, Masutaka Furue, Kenji Kabashima, Masaki Matsumoto, Keiichi I. Nakayama.

**Validation:** Akihiro Nita.

**Writing – original draft:** Akihiro Nita, Akinobu Matsumoto, Keiichi I. Nakayama.

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
