## [Decision Letter · Decision Letter 0]

24 May 2021

Dear Dr Nakayama,

Thank you very much for submitting your Research Article entitled 'The long noncoding RNA TINCR encodes a ubiquitin-like protein (ULCAR) that promotes keratinocyte proliferation and wound healing' to PLOS Genetics.

The manuscript was fully evaluated at the editorial level and by independent peer reviewers. The reviewers appreciated the attention to an important topic but identified some concerns that we ask you address in a revised manuscript

We therefore ask you to modify the manuscript according to the review recommendations. Your revisions should address the specific points made by each reviewer.

[LINK]

Yours sincerely,

Gregory S. Barsh

Editor-in-Chief

PLOS Genetics

Gregory Copenhaver

Editor-in-Chief

PLOS Genetics

Reviewer's Responses to Questions

**Comments to the Authors:**

Reviewer #1: Noncoding RNAs have attracted a lot of research interest and TINCR is one of the most well known lncRNAs after its publication in Nature. It is very important to rigorously test the concept of RNAs the exert regulatory functions without coding for proteins. Nita et al. have performed such an important test and they show that TINCR is NOT NONcoding. This result is of broad interest and important for researchers in the growing field of lncRNA biology.

The manuscript satisfies Plos Genetics’ citeria of originality, importance to researchers in the field, and broad interest to researchers in genetics and genomics. The authors’ conclusions about the functions of protein are not fully supported. For these conclusions, the methodology should be more rigorous and more substantial evidence is required for the conclusions. Alternatively, the conclusions should be phrased more cautiously, pointing to the need for further investigations.

The main open question is, by which mechanism does ULCAR accelerate the cell cycle / promote keratinocyte proliferation / wound healing? Full elucidation of the mechanism may be out of scope of this study, but mechanistic evidence and conclusions should match.

As the mechanism of cell cycle acceleration is uncertain and it is not known if this is the main role of the protein, the proposed name "Ubiquitin-Like Cell cycle AcceleratoR (ULCAR)" remains to be accepted by the research community. "ULCAR" should not be part of the title.

The title is self-contradictory (noncoding / encodes) and should be changed from

"The long noncoding RNA TINCR encodes a ubiquitin-like protein (ULCAR) that promotes keratinocyte proliferation and wound healing"

to

"A ubiquitin-like protein encoded by the reportedly “noncoding“ RNA TINCR promotes keratinocyte proliferation and wound healing"

Lines 193-195: the following sentence should be extended by stating that expression in proliferating keratinocytes is lower than in total epidermis – suggesting that expression in differentiated keratinocytes is very significant, "TINCR, and found that it was expressed not only in differentiated keratinocytes but also in actively proliferating primary mouse keratinocytes to a greater extent than in other adult mouse tissues (Fig 3F),"

Can the authors exclude or confirm that TINCR / ULCAR is expressed in hair follicles? Might its deletion affect the hair cycle and indirectly influence wound healing?

Reviewer #2: Nita et al. show that the long noncoding RNA TINCR encodes a ubiquitin-like protein (ULCAR) that promotes keratinocyte proliferation and wound healing. Together these results provide a clear and novel story about the function of ULCAR in skin regeneration. The paper is a wonderfully clear and concise description of a real breakthrough, and the experiments are presented in a straightforward and logical fashion. I could suggest more experiments or further quantitation, but I believe the authors have arrived at a correct model, with considerable novelty. The paper is suitable for Plos genetics.

Reviewer #3: This is a very exciting and novel study showing a new protein the authors termed as ULCAR was identified with the lncRNA TINCR. Using endogenous tagging, the authors convincingly demonstrated the existence of this translated protein. Overexpression of ULCAR increased cell cycle progression and suppressed differentiation in human keratinocytes, which seems not rely on TINCR RNA function. By generating ULCAR-deficient mice, the authors confirmed that loss of ULCAR results in attenuated cell cycle progression in mouse keratinocytes, and subsequently delayed recovery from skin injury. Overall this study clearly reveals a critical role for a new protein ULCAR in regulating keratinocyte differentiation and function. Given that ULCAR protein is buried within the TINCR sequence, the key is to distinguish if the observed effects are derived from ULCAR protein function or TINCR RNA function, or both. The authors did a good job in addressing this. There are several concerns that need to be clarified before publication for this novel study:

Major concerns:

1. To distinguish if the observed effects are derived from ULCAR protein function or TINCR RNA function-A good control I would suggest is to generate siRNA/shRNA/sgRNA targeting TINCR but not ULCAR, or both, and examine effects of these loss-of-function experiments.

2. The authors mentioned that in multiple cancers the abundance of TINCR is up-regulated and UCLAR negatively regulates TINCR- does TINCR also regulate ULCAR levels?

Minor concerns:

1. Fig. 1F: It seems that an N-terminal V5 tag abolishes ULCAR expression-can the authors provide some explanations?

2. Fig. 1G: The Flag signal was not observed in input but in IPs- can the authors provide some explanations?

3. Fig. 3: it is an elegant design to generate proposed mutants to disrupt either RNA or protein function- however, assays performed in this figure rely on overexpression conditions. The presence of endogenous TINCR and ULCAR may complicate the interpretation of data in this figure.

4. Fig. 3: Some assays to test if RNA structure changes generated in 3B cause deficient TINCR downstream RNA targeting would be needed.

5. It will be interesting (the authors may need to discuss this) if the authors can generate a ULCAR specific antibody to demonstrate its tissue distribution, or if its expression changes in human diseases. The former question can be answered by taking advantage of the Flag-knockin mice the authors have generated.

6. Given that this is a very novel and exciting study revealing the first protein function of ULCAR, it will be necessary to speculate how ULCAR modulates cell cycle progression? Any specific targets or binding partners of ULCAR could explain its function? Or is this related to function from other type 2 ubiquitin like molecules?

**Have all data underlying the figures and results presented in the manuscript been provided?**

Reviewer #1: Yes

Reviewer #2: Yes

Reviewer #3: Yes

PLOS authors have the option to publish the peer review history of their article (what does this mean?). If published, this will include your full peer review and any attached files.

Reviewer #1: No

Reviewer #2: **Yes: **Michele Pagano

Reviewer #3: No

---

## [Decision Letter · Decision Letter 1]

29 Jun 2021

Dear Dr Nakayama,

We are pleased to inform you that your manuscript entitled "A ubiquitin-like protein encoded by the “noncoding” RNA TINCR promotes keratinocyte proliferation and wound healing" has been editorially accepted for publication in PLOS Genetics. Congratulations!

The revised manuscript was seen by reviewers #1 and #3 of the original submission. Both reviewers recommend acceptance. There are some minor concerns remaining that we ask you address during the production process as follows:

Figures 4 and 5 need to be updated to reflect the change of the protein name from Ulcar to Tubl. Supplementary files should also be checked.

Yours sincerely,

Gregory S. Barsh

Editor-in-Chief

PLOS Genetics

Gregory Copenhaver

Editor-in-Chief

PLOS Genetics

Comments from the reviewers (if applicable):

Reviewer's Responses to Questions

**Comments to the Authors:**

Reviewer #1: Thank you for further improving the quality of the manuscript.

Reviewer #3: The authors have done an impressive work to address all my concerns raised. This is a novel and exciting study. Congratulations.

**Have all data underlying the figures and results presented in the manuscript been provided?**

Reviewer #1: Yes

Reviewer #3: Yes

PLOS authors have the option to publish the peer review history of their article (what does this mean?). If published, this will include your full peer review and any attached files.

Reviewer #1: No

Reviewer #3: No

**Data Deposition**

http://datadryad.org/submit?journalID=pgenetics&manu=PGENETICS-D-21-00588R1

**Press Queries**

---

## [Editor Report · Acceptance letter]

12 Jul 2021

PGENETICS-D-21-00588R1 

A ubiquitin-like protein encoded by the “noncoding” RNA TINCR promotes keratinocyte proliferation and wound healing 

Dear Dr Nakayama, 

We are pleased to inform you that your manuscript entitled "A ubiquitin-like protein encoded by the “noncoding” RNA TINCR promotes keratinocyte proliferation and wound healing" has been formally accepted for publication in PLOS Genetics! Your manuscript is now with our production department and you will be notified of the publication date in due course.

With kind regards,

Andrea Szabo

PLOS Genetics

On behalf of:
